# Enzymatic synthesis of lignin derivable pyridine based polyesters for the substitution of petroleum derived plastics

Alessandro Pellis [1], James W. Comerford[1], Simone Weinberger[2], Georg M. Guebitz[2,3], James H. Clark[1] & Thomas J. Farmer [1]

Following concerns over increasing global plastic pollution, interest in the production and characterization of bio-based and biodegradable alternatives is rising. In the present work, the synthesis of a series of fully bio-based alternatives based on 2,4-, 2,5-, and 2,6-pyridinedicarboxylic acid-derived polymers produced via enzymatic catalysis are reported. A similar series of aromatic-aliphatic polyesters based on diethyl-2,5-furandicarboxylate and of the petroleum-based diethyl terephthalate and diethyl isophthalate were also synthesized. Here we show that the enzymatic synthesis starting from 2,4-diethyl pyridinedicarboxylate leads to the best polymers in terms of molecular weights ($M_n = 14.3$ and $M_w$ of 32.1 kDa when combined with 1,8-octanediol) when polymerized in diphenyl ether. Polymerization in solventless conditions were also successful leading to the synthesis of bio-based oligoesters that can be further functionalized. DSC analysis show a clear similarity in the thermal behavior between 2,4-diethyl pyridinedicarboxylate and diethyl isophthalate (amorphous polymers) and between 2,5-diethyl pyridinedicarboxylate and diethyl terephthalate (crystalline polymers).

[1] The University of York, Department of Chemistry, Green Chemistry Centre of Excellence, YO10 5DD Heslington, York, UK. [2] University of Natural Resources and Life Sciences Vienna, Department of Agrobiotechnology, Institute of Environmental Biotechnology, Konrad Lorenz Strasse 20, 3430 Tulln an der Donau, Austria. [3] Austrian Centre of Industrial Biotechnology, Konrad Lorenz Strasse 20, 3430 Tulln an der Donau, Austria. Correspondence and requests for materials should be addressed to A.P. (email: alessandro.pellis@gmail.com) or to T.J.F. (email: thomas.farmer@york.ac.uk)

Society is becoming more aware of the significant problems caused by waste plastic, such as pollution of oceans and rivers[1], generation and effect of microplastics[2] on ecosystems, dependence on non-sustainable oil-based feedstocks, as well as the challenges associated with recycling a wide range of different plastics efficiently[3]. In response to this, governments and industry are combining efforts to reduce plastic-derived pollution by increasing recyclability and moving toward alternative bio-based plastics synthesized from renewable feedstocks[4]. One of the most important developments in the bio-based polymer field over the past few decades has been the work of Gandini et al. on the use of furan moieties as a sustainable replacement for the terephthalic (e.g., diethyl terephthalate, DET) and isophthalic (e.g., diethyl isophthalate, DEI) units[5]. Several multinational companies including the BASF-Avantium joint venture, Synvina[6], and Corbion[7] have shown active interest in the production and upscaling of poly(ethylene furanoate) (PEF) production, the furan equivalent of the petrol-derived poly(ethylene terephthalate) (PET).

As a result of this, a substantial number of improved methods for the chemo[8] and enzymatic[9–11] synthesis of furan-based polymers[12] and its co-polymers[13] have been published along with their corresponding mechanical, barrier[14], degradation[15], and recycling[16] properties. With such industrial investment there has also been increased intellectual property protection[17,18], especially for the production of the furandicarboxylic acid and its corresponding ester monomers (e.g., diethyl-2,5-furandicarboxylate, DEF) from sources such as 5-(hydroxymethyl)furfural (5-HMF), as well as protection of end-of-life technologies such as the use of engineered enzymes for the degradation of aromatic and aliphatic polyesters. This has driven research to focus on other bio-based monomers and polymers, such as azelaic acid and polyglycerol for applications in the cosmetic and lubricants industry[19], sorbitol-based polymers for the production of coatings[20], and itaconate containing polymers due to the post-polymerization possibilities given by the vinyl moiety[21,22]. More recently, production of 2,4-pyridinedicarboxylic acid, 2,5-pyridinedicarboxylic acid, and their mono and diester derivatives has been reported, employing a biocatalytic reaction using lignin as a substrate[23,24]. Moreover, a third pyridine derivative, 2,6-pyridinedicarboxylic acid (also known as dipicolinic acid) which composes 5–15% of the dry weight of some bacterial spores can likewise be considered as a naturally occurring compound[25,26]. All aromatic oil- and bio-derivable monomers considered in the present work are summarized in Fig. 1. Interestingly, the use of pyridine diacids in polyester synthesis seems to be a topic of remarkable inactivity and hence we see a pressing need to better understand these materials. The similarity of the pyridinedicarboxylic acids (PDCAs) to the terephthalic unit would suggest that they might offer increased rigidity if incorporated into a polymer yet retaining a potentially interesting pyridine functionality which may affect the stacking/crystallization behavior of the final product. However, this pyridine moiety is likely to also interact with conventional catalysts used for polytransesterification[27].

The focus of this paper is directed at the investigation of the biocatalyzed synthesis of PDCA-based aromatic polyesters, along with characterization of their thermal properties. A series of aromatic-aliphatic polyesters based on diethyl-2,5-furandicarboxylate and of the petroleum-based diethyl terephthalate and diethyl isophthalate are also synthesized for comparison with the pyridine-based monomers.

## Results

### Reactions in bulk
Based on previous reports on enzymatic polymerizations conducted using CaLB as biocatalyst, two different approaches were used, and herein, they will be referred to as (A) bulk reaction without solvent, until now only reported for aliphatic monomers[28,29] and (B) reactions in organic media, a traditional reaction system for the enzymatic polycondensation of aromatic moieties[30,31].

With the knowledge that enzymatic polycondensations (transesterification reactions belonging to the step-growth polymerization mechanism type) are much more efficient when activated monomers (such are diesters) are used vs dicarboxylic acids, we performed a DSC evaluation of the melting point of various dimethyl and diethyl esters (Supplementary Table 1). Bulk reaction trials at different temperatures were initially carried out as the melting points of the diethyl esters were found to be lower than the dimethyl equivalents, and the solvent-free synthesis of aliphatic polyesters, the obtained monomers conversions and molecular weights were not significantly different among the two diesters. The optimal reaction temperature for the used enzyme, lipase B from *Candida antarctica* (CaLB), was found to be 85 °C (Supplementary Table 2). These data are consistent with recent reports on CaLB-catalyzed polymerizations of aliphatic polyesters in bulk[32] and aromatic moieties in diphenyl ether (DPE)[28]. Monomers conversion rates, calculated via [1]H-NMR spectroscopy for the reactions between monomers PD24, PD25, and PD26 with $C_4$–$C_8$ aliphatic diols gave 78–82% conversion for BDO, 82–95% for HDO, and 85–93% for ODO (Fig. 2a). There is a clear trend with BDO resulting in lower monomer conversions (when compared with HDO and ODO) for each of the pyridine diesters, shown in Fig. 1. A similar trend associated with increasing carbon chain length of the diols was also observed when DET, DEI, and DEF were used as aromatic diester monomers (Fig. 2b), with BDO giving the lowest conversions amongst the used diols (78–84% vs 81–91% for HDO and ODO).

GPC analysis of the crude reaction products correlated well with monomer conversion measured by using [1]H-NMR spectroscopy. Polymerizations carried out using BDO not only show lower conversions, but were also characterized by a lower molecular weight of the obtained polymeric chains (Fig. 2c, d $M_n$, 2e, f $M_w$), Supplementary Tables 3 and 4 and Supplementary Figs. 4–9). The limited molecular masses ($M_n$ and $M_w$ > 7000 Da) obtained for all polymers using this bulk polymerization system are due to the solidification of the reaction mixture at variable times, between 8 and 28 h depending on the choice of aromatic monomer and length of the obtained polymer. These factors caused progressive hindering of the enzymatic polycondensation throughout reaction. All pyridine-based polyesters synthesized via the bulk reaction were recovered as light yellow powders (Supplementary Fig. 10).

The obtained oligomers, having a 3 < DP < 6 depending on the used pyridine diester are of high interest for the current bio-based polymers market since they could be easily used as hydrophobic cores for the synthesis of water-soluble polymers or surfactants (eg. coupling of PEG or MPEG as end cappers) or for the preparation of telechelic oligomers that can be post-polymerized (eg. crosslinked) in a 2nd reaction step to produce bio-based coatings.

### Reactions in organic media
After the successful synthesis of aromatic-aliphatic oligoesters using a solventless reaction system, the focus of the work was on the elongation of the polymeric chains since, from previous reports, enzymatic catalysts were found not to be suitable for dimethyl terephthalate, dimethyl isophthalate, 4-(2-hydroxyethoxy)benzoic methyl ester, or dimethyl-2,5-furandicarboxylate aromatic monomers. In these previous cases only short oligomers having $M_n$ ~ 400 for DMT and ~

**Fig. 1** Monomers used for the polycondensation reactions. Oil-based (top, red frame) and bio-based (bottom, green frame) monomers with related polymers

600 Da for DMI, ~ 2300 Da for 4HEBME and ~ 3300 Da for DMF were obtained using various conditions (different time, solvent and vacuum but all using CaLB as catalyst)[30,31]. Based on the above reported literature, a DPE-based solvent system was used to carry out the reactions. Temperature and vacuum were varied as reported in a recent publication[32] while in this study, monomer concentration was kept as low as 0.2 M in order to avoid a high viscosity of the reaction mixture that could hinder the enzymatic polymerization as a result of reduced mass-transfer.

The trend shown for the solventless polymerizations is preserved with the reactions between PD24, PD25, and PD26, and BDO gave conversions between 85 and 91% while using HDO and ODO as diols led to conversions ≥95% (Fig. 3a). The same trend was observed for the petroleum-derived aromatic diesters and DEF, despite the fact that the latter showed a lower conversion difference between the three diols ($C_4$, 92%; $C_6$ and $C_8$, 96%) (Fig. 3b).

Regarding molecular weights, for the reactions in DPE media, pyridine-based polymers gave higher molecular weight when compared with the polymers produced from DET, DEI, and DEF (Fig. 3c–f). The pyridine diester leading to the largest polymers was PD24 that when reacted with ODO led to polyesters with an $M_n$ of 14,300 Da and $M_w$ of 32,100 Da, respectively, giving a DP of over 50 (Fig. 3g).

The polymers produced using PD25 also gave satisfactory molecular weights ($M_n$ of 4800 Da and $M_w$ of 10,800 Da when reacted with ODO) while PD26 diester was not as successful since a maximum DP of 11 was obtained when using ODO as diol ($M_n$ = 3200 Da and $M_w$ = 7000 Da). In general, when comparing PD24 and PD25 with their petroleum-based counterpart DET and DEI, the polymers produced have higher molecular masses despite the similar isolated yields (Table 1). For the complete set of GPC data and chromatograms please refer to the

Supplementary Tables 5 and 6 and Supplementary Figs. 11–16. All polymers synthesized using the DPE-based protocol were recovered as white powders, except from the reaction between DEF and BDO that was a yellow powder (see Supplementary Figs. 17–20) with yields >60% for the polymers containing HDO and ODO as diols while lower yields (58–79%) were calculated for the polymers containing the $C_4$ diol BDO (see Supplementary Figs. 30 and 31 for [1]H-NMR conversions and yields). This differences in the recovered polymers occurred as expected since the BDO-containing polymers are more difficult to precipitate from the DPE solution; they are more soluble in MeOH due to the higher polarity of the diol component.

The MALDI analysis of the reaction products was also performed and shows how the synthesized polyesters are present mainly in their linear form, with a minor presence of the ester/ester and ester/diol end groups especially in the samples synthesized using 1.4-butanediol as the aliphatic component (Supplementary Table 12 and Supplementary Figs. 35–37).

**Polymer's thermal characterization.** From determination of the polymers' thermal properties, it is remarkable to notice that using DEI and PD24 as diesters led to more amorphous reaction products while using PD25, PD26, DET, or DEF led to the production of more crystalline polymers, independent of the aliphatic diol used (having a chain of 4, 6, or 8 carbon atoms) (Fig. 4). Similar results, in terms of more amorphous vs more crystalline polymers were observed for the reactions conducted in bulk. In polyesters the double melting peak is due to a series of possible processes: the most common is a fusion of the original crystals—recrystallization and final fusion process of the recrystallized crystals. In Fig. 4, $T_g$ and $T_m$ of the various polymers were plotted for comparison and show that while polymers synthesized

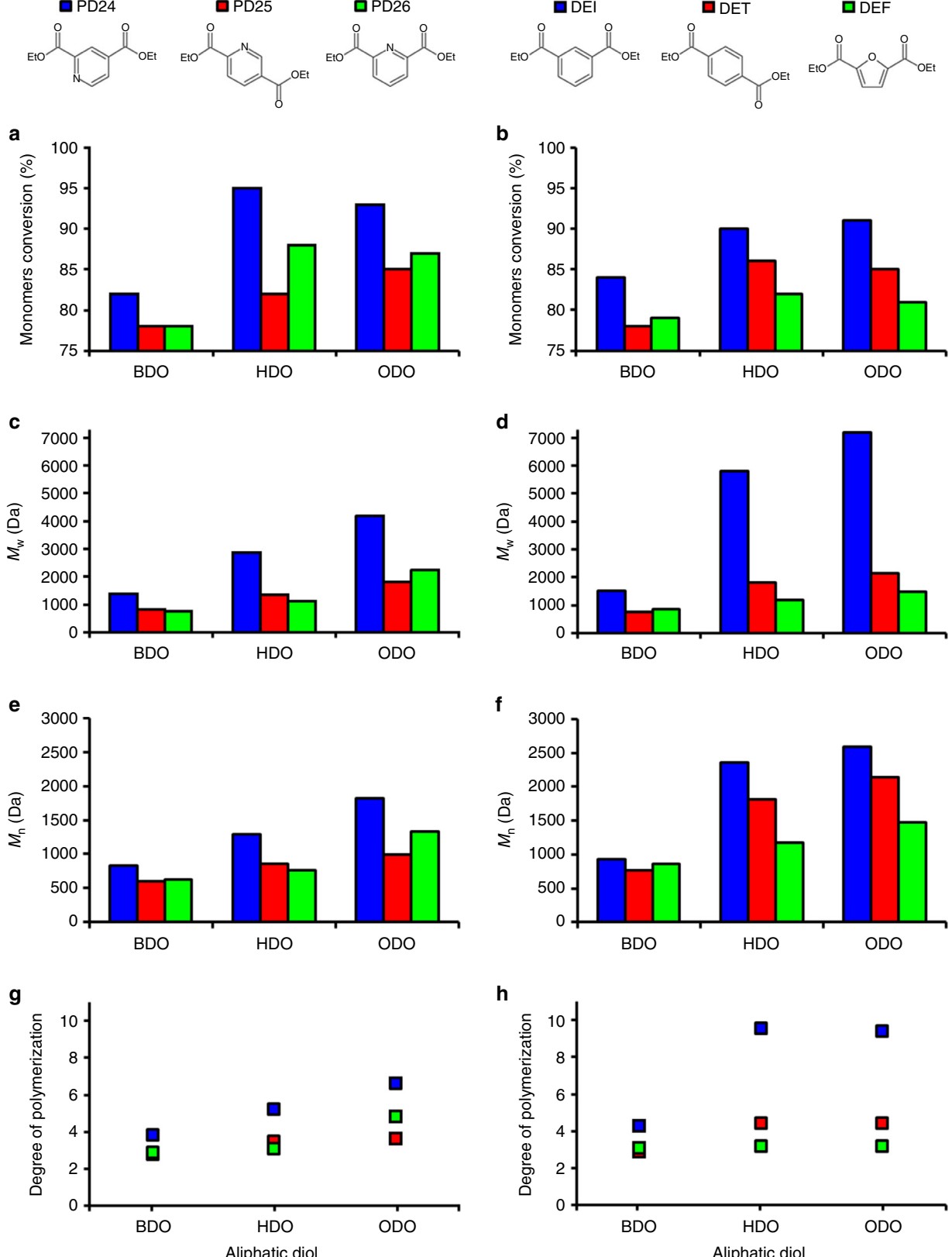

**Fig. 2** Enzymatic synthesis of pyridine-based polyesters in a solventless reaction system. **a**, **b** Monomers conversion calculated via ¹H-NMR spectroscopy; **c**, **d** number average molecular weight ($M_n$) calculated via GPC; **e**, **f** weight average molecular weight ($M_w$) calculated via GPC. **g**, **h** Degree of polymerization calculated dividing $M_n$ by the molecular mass of the polymer's constitutional repeat unit

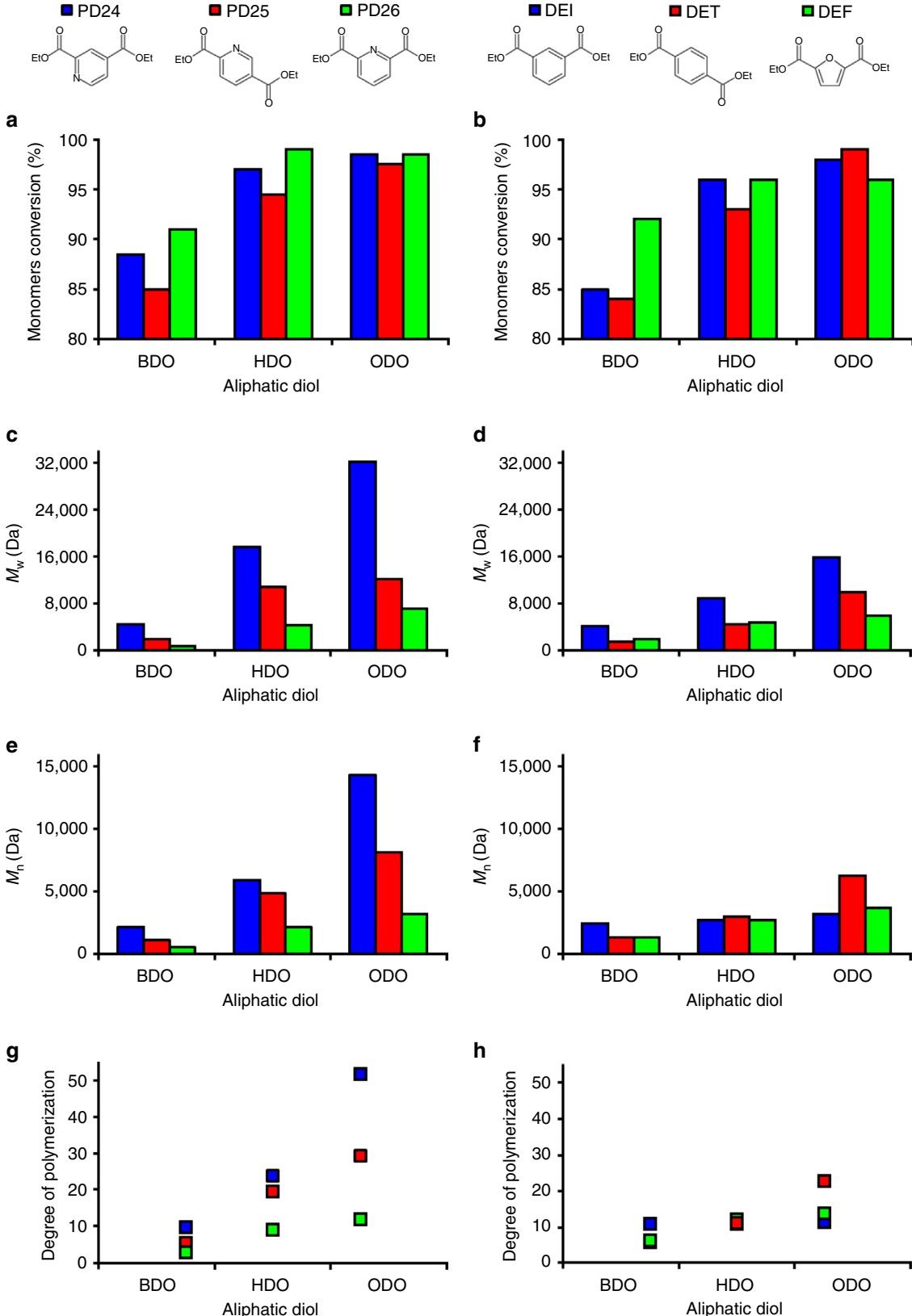

**Fig. 3** Enzymatic synthesis of pyridine-based polyesters in DPE as organic media. **a**, **b** Monomers conversion calculated via $^1$H-NMR spectroscopy; **c**, **d** number average molecular weight ($M_n$) calculated via GPC; **e**, **f** weight average molecular weight ($M_w$) calculated via GPC. **g**, **h** Degree of polymerization calculated dividing $M_n$ by the molecular mass of the polymer's repetitive unit ($M_0$)

in bulk present a similar $T_g$ of −14 vs −15 °C for poly(1,4-butylene pyridine 2,4-dicarboxylate) (pPD24-BDO) and poly(1,4-butylene isophthalate) (pDEI-BDO) and first $T_m$ values (100 vs 106 vs 100 °C for DET, PD25, and DEF, respectively) between the petro- and bio-derivable polyesters (Fig. 4a), leading to a marked difference among the synthesized polymers when the reactions are performed in DPE (Fig. 4b). This difference in $T_g$ and $T_m$ between samples appears to depend on the synthesis method. For the synthesis in bulk, the polymers solidify at different times (as discussed in the bulk polymerizations section), while the reactions in DPE allow the solubilization of the polymeric chains (including those of high molecular masses) enabling the elongation of the chain based on the enzyme's selectivity. The latter leads to polymers with high (PD24) or low (DEF, PD26) molecular weights depending on the used monomers (Fig. 3 and Supplementary Tables 8 and 10).

From the data presented in Fig. 4 it is clear that the thermal properties strongly depend on the molecular weight, especially for the polymers having lower molecular weight (oligomers synthesized in bulk). The observed trends are due to the different molecular weights: for example, pPD24-BDO synthesized in bulk has a $M_w$ of 1400 Da while pPD24-BDO synthesized in DPE has a $M_w$ of 4400 Da. This large difference can explain the substantial difference in $T_g$. These results mean that molecular weights are too low to define a constant $T_g$ value, independent of the

molecular weight. The different synthetic routes can therefore provide polymers with different molecular weights which are, characterized by different thermal behavior. Moreover, it is interesting to note that based on the monomers used, DEI and PD24 vs PD25, PD26, DEF, and DET, the polymers have more amorphous or more crystalline characteristics, independent of the diol used. These data fit very well with previous reports found in the literature. XRD analysis of selected samples was also performed (see Supplementary Figs. 32 and 33) and fully confirm the trends observed via DSC. The DSC of pPD24-ODO (see Supplementary Fig. 45) shows how weak is the energy of the $T_g$ as also confirmed in Supplementary Tables 7 and 9 where the ΔCp value is reported.

From the thermogravimetric analysis plotted in Fig. 5, a difference between the various polymers synthesized using the $C_4$, $C_6$, and $C_8$ diols in combination with PD24 (Fig. 4a), PD25 (Fig. 5b), and PD26 (Fig. 5c) is clearly seen. In particular, the polymers synthesized using 1,4-butanediol decompose at a lower temperature than the those synthesized using the $C_6$ and the $C_8$ diols. This trend is most probably due to the fact that the $C_4$ diol leads to polymers having lower molecular masses while the $C_6$ and $C_8$ diols lead to better polymers that are more thermally stable. In addition, 1,4-butanediol might be converted to volatile tetrahydrofuran during the pyrolysis of the sample as it is well known that weak end groups are affecting the polymer degradation[33]. This thermogravimetric data agree with trends previously observed for the polymerizations of various aliphatic polyesters in bulk reaction systems using CaLB as the biocatalyst[30].

If we take the polymers synthesized in DPE using DEI as the diester together with the various diols, we can observe that there is not a significant difference between the obtained molecular masses (Fig. 3). This is fully reflected in the TGA profiles of the samples (see ESI, Figs. S21–24) that show similar Td₅₀ values amongst the various polymers ($C_4$ = 372 °C, $C_6$ = 379 °C, and $C_8$ = 380 °C) (for the complete set of TGA data see Supplementary Table 11).

**Separation of pyridine-based polymers for recycling purposes.** Knowing that the separation of plastics for recycling purposes is based mainly on near infrared spectroscopy (NIR)[34], we compared the polymers synthesized from the petro-derived monomers DET and DEI with the bio-derivable polyesters obtained

**Table 1 Yields of the enzymatic synthesis reactions in DPE after the precipitation and purification steps**

| Diester | Diol | Yield [%] | Diester | Diol | Yield [%] |
|---------|------|-----------|---------|------|-----------|
| DET | BDO | 71 | PD24 | BDO | 58 |
| | HDO | 90 | | HDO | 68 |
| | ODO | 96 | | ODO | 72 |
| DEI | BDO | 49 | PD25 | BDO | 79 |
| | HDO | 64 | | HDO | 88 |
| | ODO | 66 | | ODO | 89 |
| DEF | BDO | 69 | PD26 | BDO | 92 |
| | HDO | 93 | | HDO | 89 |
| | ODO | 92 | | ODO | 90 |

Yields were calculated dividing the recovered amount of polymer (mg) by the total theoretical amount of polymer that should be produced when all the reagent in defect is consumed (mg)

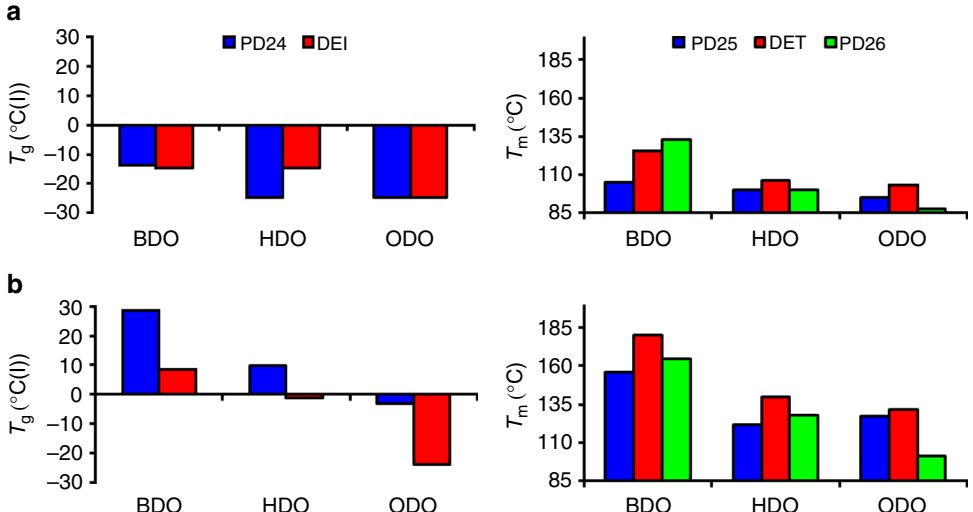

**Fig. 4** DSC analysis of the polymers. Glass transition temperatures ($T_g$) and melting temperatures ($T_m$) of the polymers synthesized in (**a**) bulk and (**b**) diphenyl ether

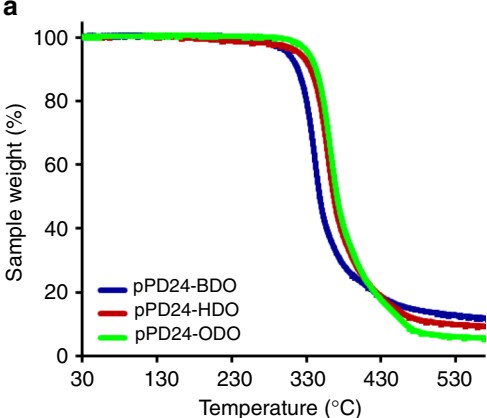

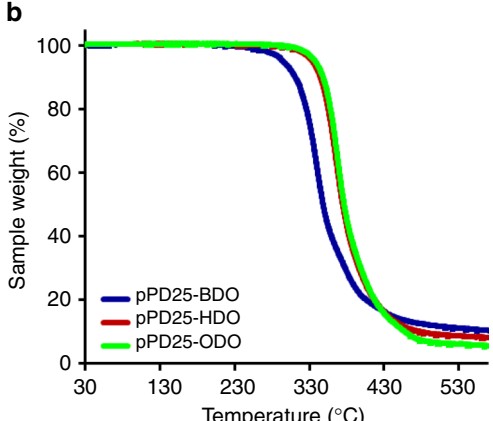

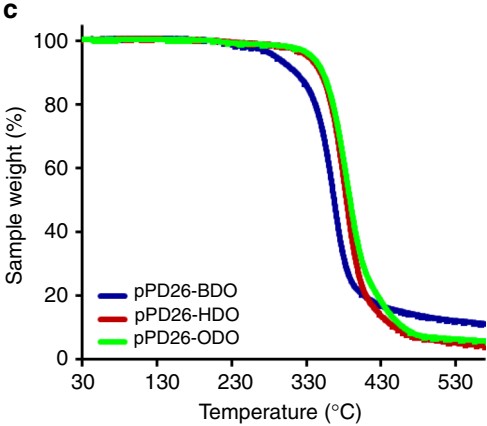

**Fig. 5** Thermogravimetric analysis of the polymers. **a** PD24, **b** PD25, and **c** PD26 in combination with 1,4-butanediol ($C_4$, blue line), 1,6-hexanediol ($C_6$, red line), and 1,8-octanediol ($C_8$, green line) in the DPE reaction system using immobilized CaLB as the biocatalyst. Runs were conducted using $N_2$ as the inert gas

from DEF (furan-based) and PD24, PD25, and PD26 (pyridine-based). Interestingly, using FT-IR, polymers containing DET and PD25 are in our case easily distinguishable by observing the $1614–1548 \, cm^{-1}$ region, where polymers containing DET have a single representative peak at $1578 \, cm^{-1}$ (see Supplementary Fig. 25) and polymers containing PD25 have a characteristic double peak at 1594 and $1571 \, cm^{-1}$ (see Supplementary Fig. 26). This difference between DET and PD25 is well-preserved throughout the synthesized samples and appears not to be related to the diol used (see Supplementary Fig. 27 for comparison

between polymers synthesized from PD25 and HDO or ODO). For distinguishing the polymers obtained from DEF and PD26, we cannot use the peak at $\sim 1570 \, cm^{-1}$ because the difference in absorption between the two aromatic moieties is limited (1573 vs $1578 \, cm^{-1}$). Fortunately, the characteristic C=O peak at ~1700 for all polymers containing PD26 has a consistent "shoulder" (1738 and $1717 \, cm^{-1}$) while for DEF a single maximum at $1722 \, cm^{-1}$ was observed (see Supplementary Fig. 28). A clear distinction between the polymers synthesized from DEI and PD24 is instead rather difficult since they both present a single C=O peak at around $1720–1722 \, cm^{-1}$ and a double peak in the $1600/1560 \, cm^{-1}$ region (see Supplementary Fig. 29). This lack of separation clarity between polymers containing DEI and PD24 is not critical since at present as DEI is typically not used in the industrial production of polyesters.

## Discussion

The present work sheds light on the possibility of synthesizing 100% bio-based polyesters from several pyridine derivatives such as 2,4-diethyl pyridinedicarboxylate (PD24), 2,5-diethyl pyridinedicarboxylate (PD25), and 2,6-diethyl pyridinedicarboxylate (PD26) via enzymatic catalysis using solventless or diphenyl ether-based reaction systems. While the reactions conducted in bulk lead to oligomers with $3 < DP < 6$ and $M_n < 7000$, the reactions conducted in DPE lead to polymers with $M_n > 14000$ (DP > 50) doubling the $M_n$ of the equivalent polymers synthesized from petroleum-based derivatives and four times higher than the polymers produced using 2,5-diethyl furandicarboxylate as the diester monomer. The thermal behavior of the obtained polymers was also investigated and strict correlations between the DEI and PD24 and the DET and PD25 diesters were observed in terms of obtained molecular weights, yields and obtained crystallinity. Furthermore, the synthesized polymers are easily distinguishable from the classic terephthalate polyesters waste using near IR spectroscopy as currently used for separation from plastic waste streams for recycling purposes.

## Methods

**Chemicals and enzymes**. Diethyl terephthalate (DET) was purchased from TCI. Diethyl isophthalate (DEI) was purchased from Syntree Inc. Diethyl-2,5-furandicarboxylate (DF25) and diethyl pyridine-2,4-dicarboxylate (PD24) were purchased from Carbosynth. Diethyl pyridine-2,5-dicarboxylate (PD25) and diethyl pyridine-2,6-dicarboxylate (PD26) were purchased from TCI and are not yet available from bioderived sources. 1,8-octanediol (ODO) was purchased from Acros Organics. 1,4-butanediol (BDO), 1,6-hexanediol (HDO), diphenyl ether (DPE), and all other chemicals and solvents were purchased from Sigma-Aldrich and used as received if not otherwise specified.

*Candida antarctica* lipase B (CaLB) immobilized onto acrylic resin (iCaLB) was purchased from Sigma-Aldrich (product code L4777). The enzyme was vacuum dried for 96 h and stored in a desiccator before use.

**Enzymatic solventless polymerizations**. Six millimoles of diester and 6 mmol of diol (diester:diol ratio = 1:1) were added in a 25-mL round bottom flask. The mixture was then stirred at 85 °C until a homogeneous melt was obtained. In total, $10\% \, w \, w^{-1}$ (calculated on the total amount of the monomers) of iCaLB was then added and the reaction run for 6 h at 1000 mbar. A vacuum of 20 mbar was subsequently applied for additional 90 h maintaining the reaction temperature at 85 °C. With progression of the reaction the mixture became solid at different times therefore hindering polymerization. The product was recovered by adding warm chloroform to the reaction mixture. The biocatalyst was then removed via a filtration step and the solvent evaporated under vacuum. The reactions resulted in formation of an of white/yellow powdery polymer.

**Enzymatic polymerizations in organic media**. In total, $8 \times 10^{-4}$ mol of diester (0.2 M) and $8 \times 10^{-4}$ mol of BDO (0.2 M) (diester:diol ratio = 1:1) were added together with 4 mL of DPE in a 25-mL round bottom flask. The mixture was then stirred at 85 °C until complete dissolution of the monomers in the solvent. In total, $10\% \, w \, w^{-1}$ (calculated on the total amount of the monomers) of iCaLB was then added and the reaction was run for 6 h at 1000 mbar. A vacuum of 20 mbar was subsequently applied for an additional 90 h while maintaining the reaction $T$ at 85 °C. Warm chloroform was added to the reaction mixture to solubilize the

polymer product and the biocatalyst was filtered off. The chloroform was then removed under vacuum. The polymer-DPE mixture was subsequently crashed out in ice-cold methanol achieving precipitation of the products. Three methanol washing steps were subsequently performed in order to remove the residual DPE. The reactions led to white powdery polymerization products (see Supplementary Figs. 17–20). See Supplementary Figs. 1–3 for the $^1$H-NMR spectroscopy data of the various purification steps. The reproducibility of the reactions was certified by performing duplicate runs of selected reactions and all relative characterization data are included in the ESI file.

**Nuclear magnetic resonance (NMR) spectroscopy**. $^1$H and $^{13}$C-NMR spectroscopy analysis were performed on a JEOL JNM-ECS400A spectrometer at a frequency of 400 MHz for $^1$H and 100 MHz for $^{13}$C. CDCl$_3$ was used as NMR solvent if not otherwise specified. A drop of TFA was added to ensure the full solubilization of the polyDET-BDO synthesized in DPE. $^1$H-NMR analysis was performed on all synthesized samples (see Supplementary Figs. 38–43 for the fully assigned NMR spectra of some selected reactions carried out in DPE).

**Gel permeation chromatography (GPC)**. Samples were dissolved in CHCl$_3$ and filtered through a cotton filter prior to passing into a HPLC vial. Gel permeation chromatography was carried out at 30 °C on an Agilent Technologies HPLC System (Agilent Technologies 1260 Infinity) connected to a 17369 6.0 mm ID × 40 mm L HHR-H, 5 µm Guard column and a 18055 7.8 mm ID × 300 mm L GMHHR-N, 5 µm TSKgel liquid chromatography column (Tosoh Bioscience, Tessenderlo, Belgium) using 1 mL min$^{-1}$ CHCl$_3$ as mobile phase. A drop of TFA was added to ensure full solubilization of the polyDET-BDO synthesized in DPE. An Agilent Technologies G1362A refractive index detector was employed for detection. The molecular weights of the polymers were calculated using linear polystyrene calibration standards. GPC tracks of the polymers having the lowest molecular masses (polyDET-BDO and polyDEF-BDO) have a higher error due to the slightly higher baseline fluctuation but analysis are still fully reproducible and the results are significant.

**Matrix assisted laser desorption ionization (MALDI)**. MALDI-TOF MS analysis were carried out by using a Bruker Solarix-XR FTICR mass spectrometer and the relative software package for the acquisition and the processing of the data. An acceleration voltage of 25 kV, using DCTB as matrix and KTFA as ionization agent were used. Ten microliters of sample were mixed with 10 µL of matrix solution (40 mg mL$^{-1}$ DCTB in THF) and 3 µL of KTFA (5 mg mL$^{-1}$). In total, 0.3 µL of the mixture were applied on the plate and the measurement was conducted in positive mode with the detector set in reflector mode. All TGA data were summarized in Supplementary Table 12 and Supplementary Figs. 35–37.

**Differential scanning calorimetry (DSC)**. DSC experiments were performed on a TA Instruments Q2000 DSC under an inert gas atmosphere (N$_2$). The used heating and cooling rates were set to 5 °C over the $T$ range of −60–200 °C. Sample mass was of between 5 and 10 mg for all measured samples. The $T_g$ values were reported from the second heating scan. The melting points of the starting compounds were also determined and are reported in Supplementary Table 1 and additional DSC thermograms were added as Supplementary Figs. 44–46.

**Thermogravimetric analysis (TGA)**. TGA was performed on a PL Thermal Sciences STA 625 thermal analyzer. Approximately 10 mg of accurately weighed sample in an aluminum sample cup was placed into the furnace with a N$_2$ flow of 100 mL min$^{-1}$ and heated from room temperature to 625 °C at a heating rate of 10 °C min$^{-1}$. From the TGA profiles the temperatures at 10 and 50% mass loss (TD$_{10}$ and TD$_{50}$, respectively) were subsequently determined. All TGA data were summarized in Supplementary Table 11 and Supplementary Figs. 21–24.

**Fourier transformation infrared spectroscopy**. Fourier transformation infrared spectroscopy (FT-IR) analysis of the synthesized polymers was performed on a PerkinElmer 400 spectrometer using the attenuation total reflectance setting. The same pressure was applied on the outer surface of all analyzed samples. A number of 32 scans were recorded using a 1 cm$^{-1}$ resolution. All spectra were processed using the automated baseline correction and the data auto-tune functions (see Supplementary Figs. 25–29).

**X-ray diffractometry**. The samples were measured between 5 and 52° 2q on a Panalytical Empyrean X-ray diffractometer equipped with Co Ka (1.790307 Å) radiation. They were mounted on a zero-background offcut Si holder and a beam knife used to reduce the air scatter at low angle.

See Supplementary Methods for all performed computational analysis and Supplementary Fig. 34.

## Data availability

All data used in the preparation of this manuscript is contained within this document, the electronic supplementary information, or available on request from https://doi.org/10.15124/57ad733f-3f13-423d-af27-dbc0c3880f11.

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

## Acknowledgements

Alessandro Pellis thanks the FWF Erwin Schrödinger fellowship (grant agreement J 4014-N34) for financial support. James Clark, Thomas Farmer, and James Comerford thank the Biotechnology and Biological Sciences Research Council (BBSRC, grant BB/N023595/1) for funding their involvement in this research. The authors thank Biome bioplastics for financial support including supplying the monomers used in this study. Molecular volumes and sigma-surfaces are courtesy of, and authorized for the purposes of, Circa Group Pty Ltd. All authors would like to thank Dr. David Walker from Warwick University for the help with the XRD analysis.

## Author contributions

A.P. performed the enzymatic polymer synthesis and the polymer characterizations. J.W.C. helped with the thermal analysis of the polymers. S.W. performed the GPC analysis of the polymers. A.P. planned the experiments and wrote the manuscript. J.H.C. and T.J.F. supervised the work. G.M.G. and J.H.C. contributed with discussions and manuscript revisions. All authors corrected the manuscript and discussed the data prior to submission.

## Additional information

**Competing interests:** The authors declare no competing interests.

