## [Peer Review File · Nature Communications]

Reviewers' comments:

Reviewer #2 (Remarks to the Author):

The manuscript contains a very important research work in terms of green polymer chemistry leading to very interesting potentially applicable materials.

The manuscript should be published in Nature Communications.

I have a few tips to improve the manuscript:

- did the authors perform X-ray diffraction studies of their materials? It might be good to compare it to their DSC data.
- do the authors know what end-groups their oligomers have? And if their end-group fidelity is good or if there is a large variety?
- why is the GPC of BDO so bumpy in all cases? Is there a reason to be found in the step-growth process?
- the produced materials are thermally not very stable and the TGA curve goes to 0%. Usually such structures have a higher decomposition temperature (even as oligomers) and have more than 10% remaining. Do the authors know the reason for this?

Reviewer #3 (Remarks to the Author):

The paper describes an original synthesis of polyesters in which monomers of potential natural origin are proposed. The scientific and technological context addressed is clearly stated and it influences a wide field of researchers and developers of new materials.

The following remarks can be of help to make the paper stronger and more convincing and, consequently, ready to be accepted for publication.

General Remarks

- The monomers proposed are not yet available from natural sources, therefore the authors use their chemically synthesized equivalents. This point should be discussed in the introduction, in order to convince the reader further on the impact of the proposed work for the bio-based economy.
- While the authors properly discuss and perform experiments in view of the materials recyclability, they should also comment on the degradability and the possible toxicity of the degradation products.
- A statistical analysis has not been duly performed in the paper, so it is difficult to understand if the trends detected and discussed (e.g. in terms of the effect of the type of the aliphatic diol used on the monomer conversion, degree of polymerisation, molecular weight)
- The computational studies are hardly belonging to the rest of the paper, since no clear indication from this work is used either to design further experiments or to their interpretation. Therefore, it is suggested to remove this part or to include it in the supplementary material.

Specific Remarks

- Even in the "greener" solventless polymerization procedure, the authors recover the products with chloroform. Are there other, more environmentally friendly alternatives available, in view of the upscaling of the process to larger scales?
- The part described on page 11, from line 209 and including Scheme 2, does not belong to the paper, since it is only a potential development of the technology described, without any experimental proof of concept being carried out. Therefore, it should be removed or briefly summarised in the conclusion section.
- The sentence on page 15, from line 269, should be rewritten in a more clear English. Moreover some experimental proof should be provided to support the statement.
- On page 20-21, from line 352 authors probably meant "from 2 and 4" instead of "from 1 and 4".

Point-by-point answer to the reviewer's comments

Reviewer #2 (Remarks to the Author):

The manuscript contains a very important research work in terms of green polymer chemistry leading to very interesting potentially applicable materials.

The manuscript should be published in Nature Communications.

Answer 1: we thank the reviewer for the positive evaluation of our manuscript and for suggesting its publication in Nature Communications.

I have a few tips to improve the manuscript:

- did the authors perform X-ray diffraction studies of their materials? It might be good to compare it to their DSC data.

Answer 2: XRD analysis of selected samples was performed and a new section was added in the main text after the DSC discussion (see above).

- do the authors know what end-groups their oligomers have? And if their end-group fidelity is good or if there is a large variety?

Answer 3: MALDI analysis was performed to determine the end-groups of the synthesized polymers. The results show how there is a high fidelity for all synthesized PDCA-based polyesters that are mainly represented in the cyclic form. Some ester/ester and ester/diols end groups were also found in some samples but results in line with previous reports on MALDI end-group analysis of polyesters. A comment on that was added in the main text and a Table detailing the results was added in the ESI file (Table S12 and Figures S35-37)

- why is the GPC of BDO so bumpy in all cases? Is there a reason to be found in the step-growth process?

Answer 4: see comment above.

- the produced materials are thermally not very stable and the TGA curve goes to 0%. Usually such structures have a higher decomposition temperature (even as oligomers) and have more than 10% remaining. Do the authors know the reason for this?

Answer 5: we thank the reviewer for the precise analysis of our TGA data that matches very well our observations. Unfortunately, we don't have an explanation for this phenomenon: all polymers synthesized from the petroleum and furan-based diesters go to 0% while all the pyridine-based ones reach a maximum, as the reviewer correctly noticed, around 10%. We believe that this difference is due to the different source of the used monomers (petrol-based vs bio-based) that leads to different leftover components.

Reviewer #3 (Remarks to the Author):

The paper describes an original synthesis of polyesters in which monomers of potential natural origin are proposed. The scientific and technological context addressed is clearly stated and it influences a wide field of researchers and developers of new materials.

The following remarks can be of help to make the paper stronger and more convincing and, consequently, ready to be accepted for publication.

Answer 1: we thank the reviewer for the positive evaluation of our manuscript and for the given suggestions to improve it.

General Remarks

- The monomers proposed are not yet available from natural sources, therefore the authors use their chemically synthesized equivalents. This point should be discussed in the introduction, in order to convince the reader further on the impact of the proposed work for the bio-based economy.

Answer 2: the author is right, a statement was added in the materials and methods section and the relative section now reads: "Diethyl pyridine-2,5-dicarboxylate (PD25) and diethyl pyridine-2,6-dicarboxylate (PD26) were purchased from TCI and are not yet available from bioderived sources."

- While the authors properly discuss and perform experiments in view of the materials recyclability, they should also comment on the degradability and the possible toxicity of the degradation products.

Answer 3: degradability studies in terms of *in vitro* enzymatic hydrolysis and standard biodegradation testing both in water and soil are currently ongoing and will be published in timely manner since the biodegradation results will take several months to be completed. This work will be presented as a follow-up paper since the upscaling of the synthesis procedures (up to kilo scale) have also been investigated.

- A statistical analysis has not been duly performed in the paper, so it is difficult to understand if the trends detected and discussed (e.g. in terms of the effect of the type of the aliphatic diol used on the monomer conversion, degree of polymerisation, molecular weight)

Answer 4: as it is possible to observe from the data in the ESI file, all reactions relative to this paper were performed in duplicates for the sake of reproducibility and lead to very similar values in terms of conversions, DP, molecular weights, etc. In order to convince the reviewer on the validity of the presented results, average and standard deviations were plotted for some of the samples in the ESI file and now look as:

Figure S30. Monomers conversion calculated via $^1\text{H-NMR}$ (average \pm standard deviation) for the reactions between the pyridine diesters and the various aliphatic diols.

Figure S31. Calculated yield and comparison between the PD24, DEI and PD26 aromatic diesters and the various aliphatic diols (average \pm standard deviation).

- The computational studies are hardly belonging to the rest of the paper, since no clear indication from this work is used either to design further experiments or to their interpretation. Therefore, it is suggested to remove this part or to include it in the supplementary material.

Answer 5: as recommended, the computational part of the work was moved in the ESI.

Specific Remarks

- Even in the "greener" solventless polymerization procedure, the authors recover the products with chloroform. Are there other, more environmentally friendly alternatives available, in view of the upscaling of the process to larger scales?

Answer 6: we are currently investigating some alternative green solvents for both the solvent-based and the bulk processes. In detail, we are now exploring Cyrene as potential substitute for diphenyl ether and 2,2,5,5-tetramethyloxolane as potential substitute for the chloroform used during the work-up procedure. We just started this part of the work and we do not have any data ready for being included in the present publication..

- The part described on page 11, from line 209 and including Scheme 2, does not belong to the paper, since it is only a potential development of the technology described, without any experimental proof of concept being carried out. Therefore, it should be removed or briefly summarised in the conclusion section.

Answer 7: the relative section of the paper was shortened and the scheme removed from the main text as suggested.

- The sentence on page 15, from line 269, should be rewritten in a more clear English. Moreover some experimental proof should be provided to support the statement.

Answer 8: following also the comments from Reviewer 1, the sentence was corrected accordingly and now reads as: "In polyesters the double melting peak is due to a series of possible processes: the most common is a fusion of the original crystals-recrystallization and final fusion process of the recrystallized crystals. In Figure 4, T_g and T_m of the various polymers were plotted for comparison."

- On page 20-21, from line 352 authors probably meant "from 2 and 4" instead of "from 1 and 4".

Answer 9: we have corrected the mistake and the sentence now reads: "This lack of separation clarity between polymers containing DEI and PD24 is anyhow not critical since at present DEI is typically not used in the industrial production of polyesters."

Reviewer #2 (Remarks to the Author):

The authors answered all raised questions perfectly and changed the manuscript accordingly.

I strongly support publication of the current version.

Reviewer #3 (Remarks to the Author):

The authors have significantly improved the paper following the reviewers' suggestions. In my opinion, the paper may now be accepted for publication.

Point-by-point answer to the reviewer's comments

Reviewer #2 (Remarks to the Author):

The authors answered all raised questions perfectly and changed the manuscript accordingly.

I strongly support publication of the current version.

Answer1: we thank again the reviewer for the constructive comments provided and for availing the publication of our manuscript.

Reviewer #3 (Remarks to the Author):

The authors have significantly improved the paper following the reviewers' suggestions. In my opinion, the paper may now be accepted for publication.

Answer1: we thank again the reviewer for the constructive comments provided and for availing the publication of our manuscript.

REVIEWERS' COMMENTS:

Reviewer #2 (Remarks to the Author):

The editor asked me to look at the reply of the authors on the points raised by reviewer #1 after I already confirmed that my questions and concerns were answered perfectly.

In the current version the authors answered all points raised by reviewer #1 in a very tedious fashion. I consider some of the points raised not very valuable (quality of baselines etc.) but they were still answered in a convincing way. Even a repository will be created for NMR spectra of all compounds synthesized.

I would like to strongly advise to accept this manuscript for publication in Nature Communications.

Point-by-point answer to the reviewer's comments

Reviewer #2 (Remarks to the Author):

The editor asked me to look at the reply of the authors on the points raised by reviewer #1 after I already confirmed that my questions and concerns were answered perfectly.

In the current version the authors answered all points raised by reviewer #1 in a very tedious fashion. I consider some of the points raised not very valuable (quality of baselines etc.) but they were still answered in a convincing way. Even a repository will be created for NMR spectra of all compounds synthesized.

I would like to strongly advise to accept this manuscript for publication in Nature Communications. In the comment suitable for publication in Nature Communications.

Answer: we thank again the reviewer for the constructive comments provided and for the positive evaluation of our manuscript.